# Exposure to Low Zearalenone Doses and Changes in the Homeostasis and Concentrations of Endogenous Hormones in Selected Steroid-Sensitive Tissues in Pre-Pubertal Gilts

**DOI:** 10.3390/toxins14110790

**Published:** 2022-11-11

**Authors:** Magdalena Gajęcka, Łukasz Zielonka, Andrzej Babuchowski, Maciej Tadeusz Gajęcki

**Affiliations:** 1Department of Veterinary Prevention and Feed Hygiene, Faculty of Veterinary Medicine, University of Warmia and Mazury in Olsztyn, Oczapowskiego 13, 10-718 Olsztyn, Poland; 2Dairy Industry Innovation Institute Ltd., Kormoranów 1, 11-700 Mrągowo, Poland

**Keywords:** zearalenone, low doses, gonads, hypothalamus, pituitary gland, steroid hormones, pre-pubertal gilts

## Abstract

This study was undertaken to analyze whether prolonged exposure to low-dose zearalenone (ZEN) mycotoxicosis affects the concentrations of ZEN, α-zearalenol (α-ZEL), and β-zearalenol (β-ZEL) in selected reproductive system tissues (ovaries, uterine horn—ovarian and uterine sections, and the middle part of the cervix), the hypothalamus, and pituitary gland, or the concentrations of selected steroid hormones in pre-pubertal gilts. For 42 days, gilts were administered per os different ZEN doses (MABEL dose [5 µg/kg BW], the highest NOAEL dose [10 µg/kg BW], and the lowest LOAEL dose [15 µg/kg BW]). Tissue samples were collected on days seven, twenty-one, and forty-two of exposure to ZEN (exposure days D1, D2, and D3, respectively). Blood for the analyses of estradiol and progesterone concentrations was collected in vivo on six dates at seven-day intervals (on analytical dates D1–D6). The analyses revealed that both ZEN and its metabolites were accumulated in the examined tissues. On successive analytical dates, the rate of mycotoxin accumulation in the studied tissues decreased gradually by 50% and proportionally to the administered ZEN dose. A hierarchical visualization revealed that values of the carry-over factor (CF) were highest on exposure day D2. In most groups and on most exposure days, the highest CF values were found in the middle part of the cervix, followed by the ovaries, both sections of the uterine horn, and the hypothalamus. These results suggest that ZEN, α-ZEL, and β-ZEL were deposited in all analyzed tissues despite exposure to very low ZEN doses. The presence of these undesirable compounds in the examined tissues can inhibit the somatic development of the reproductive system and compromise neuroendocrine coordination of reproductive competence in pre-pubertal gilts.

## 1. Introduction

Raw materials and feed components of plant origin are often contaminated with undesirable substances such as mycotoxins [1], which pose a health risk to humans [2] and various livestock species, pigs in particular [3]. The symptoms and health (toxicological) risks associated with exposure to high doses of these compounds have been investigated with regard to a limited number of mycotoxins, including ZEN and its metabolites: α-ZEL and β-ZEL [4,5,6,7]. In light of the hormesis paradigm, which posits that low doses of undesirable substances exert beneficial effects on the body [8,9], the consequences of prolonged exposure to low concentrations of mycotoxins (which are frequently found in animal feed) should be studied. Numerous studies involving mammals have been undertaken to identify potential physiological dysfunctions resulting from exposure to the pure parent compound [10,11,12,13,14,15] without metabolites, or to modified mycotoxins [9], particularly in the reproductive [16] and hormonal systems [10].

The existing research indicates that exposure to low ZEN concentration may cause side effects that are hard to predict [17]. The observed changes are influenced by the administered dose and duration of exposure [18]. Low mycotoxin doses can elicit surprising effects: for example, the body’s failure to detect undesirable substances such as mycotoxins [19]. Long-term exposure to orally administered ZEN leads to increased mycotoxin accumulation in target cells [13,14] and induces the compensatory effect [20] by altering the analyzed indicators, for example, in the reproductive system [21], changing the activity of the hypothalamic-pituitary-gonadal (HPG) axis [22,23] and disrupting hormonal homeostasis in pre-pubertal animals. Homeostasis is restored [24] in subsequent stages of exposure [10]. These factors, as well as the promiscuity [25] of ZEN and its known metabolites [17], and the type and intensity of physiological reaction in gilts exposed to this mycotoxin, point to the need for further study into the effects of low dietary zearalenone doses. 

Based on the results of our prior studies [10,14,15], a low ZEN dose was defined by examining whether clinical symptoms of ongoing mycotoxicosis were present. Three ZEN doses were proposed based on our previous work and a review of the literature: (i) the lowest dose which elicits clinical symptoms [3] (>10 μg ZEN/kg of body weight, BW), defined as the lowest observed adverse effect level (LOAEL) [19]; (ii) the highest dose which does not elicit clinical symptoms (subclinical states) (= 10 μg ZEN/kg BW), defined as the no observed adverse effect level (NOAEL) [26]; and (iii) the lowest measurable dose which enters into positive interactions with the host organism in different stages of life (<10 μg ZEN/kg BW), defined as the minimal anticipated biological effect level (MABEL) [6,27,28].

Since zearalenone is a mycoestrogen, the dose-reaction paradigm has been subverted and replaced with the low dose hypothesis [17]. This applies, in particular, to hormonally active chemical compounds [29]. The ambiguous dose-response relationship prevents a direct, monotonic extrapolation or meta-analysis of the risks (including clinical symptoms and the results of laboratory analyses) associated with the transition from a high to a low dose [3,24]. On the other hand, ZEN’s toxicity can be attributed to its chemical structure and ability to interact with steroid hormone receptors in many internal organs [30]. Zearalenone can also cross the barrier between the cerebral capillary blood and the interstitial fluid of the brain and affects neurons in the central nervous system [31,32]. Recent research has shown that exposure to ZEN disturbs the synthesis of neuronal factors and enzymes in brain neurons. In pigs (including pre-pubertal gilts), reproductive functions are controlled by complex regulatory networks which integrate peripheral and internal signals, thus affecting brain regions that control, e.g., the HPG axis [33]. By binding to specific receptors on gonadotropic cells in the pituitary gland, ZEN and/or its metabolites block these signals, which inhibits the biosynthesis and release of two gonadotropins—the luteinizing hormone (LH) and the follicle-stimulating hormone (FSH) [23]—and decreases the amplitude of LH pulsation [34]. LH and FSH are essential for gonadal development and fertility, and they bind to gonadal receptors to regulate gametogenesis and steroidogenesis [35,36].

Based on the above observations and a review of the literature, we hypothesized that ZEN present in feed materials at very low concentrations is accumulated in the reproductive systems, hypothalamus, and pituitary glands of pre-pubertal gilts. Thus, the aim of this study was to determine whether exposure to low doses of zearalenone (MABEL dose [5 μg/kg BW], the highest NOAEL dose [10 μg/kg BW], and the lowest LOAEL dose [15 μg/kg BW]) administered per os to sexually immature gilts over a period of 42 days affects the levels of zearalenone, alfa-ZEL, and beta-ZEL in selected reproductive system tissues (ovaries, uterine horn—ovarian and uterine sections, and the middle part of the cervix), the hypothalamus and pituitary gland, along with whether it affects the peripheral blood levels of two steroid hormones: estradiol and progesterone.

## 2. Results 

The presented results were obtained as part of a large-scale experiment which did not reveal clinical signs of ZEN mycotoxicosis. However, differences were frequently observed in the values of the carry-over factor (CF) of zearalenone and its metabolites in intestinal tissues, in CYP1A1 and GSTπ1 expression in the large intestine, in selected serum biochemical profiles, in the myocardium and the coronary artery, in cecal water genotoxicity, in selected steroid concentrations, in intestinal microbiota parameters, and in the weight gain of animals. Samples gathered from the same animals were analyzed. Previous findings were published in several publications [10,11,12,13,14,15,17,37].

### 2.1. Experimental Feed

Experimental diets did not contain any mycotoxins, or their amounts were below the limit of detection (LOD). The concentrations of masked and/or modified mycotoxins were not determined.

### 2.2. Results of Laboratory Analyses 

#### 2.2.1. Concentrations of ZEN and Its Metabolites in Selected Tissues 

The CF values of ZEN and its metabolites differed considerably (see Table 1, Table 2 and Table 3, Figure 1, Figure 2 and Figure 3, Appendix A) not only on different exposure dates, but also between groups and the analyzed tissues. These differences are evident in the resulting tree maps, where hierarchical data are presented by a series of nested rectangles (see Appendix A). 

In all groups, mean ZEN concentrations decreased in all analyzed tissues on successive exposure dates (see Table 1). In group ZEN5 (MABEL dose), significant differences were found in the uterine horn (ovarian and uterine sections) and the middle part of the cervix on D2 and D3 relative to D1. In group ZEN10 (NOAEL dose), significant differences were observed in the uterine section of the uterine horn on D2 and D3, and in the ovarian section of the uterine horn only on D2, in comparison to D1. In group ZEN15 (LOAEL dose), significant differences were noted in the ovarian section of the uterine horn on D2 and D3 vs. D1, and in the middle part of the cervix on D3 vs. D1.

The statistical analysis of ZEN concentrations (see Table 1) in the analyzed tissues, on different exposure dates and in different groups, revealed that ZEN levels increased with a rise in the administered ZEN dose. On D1, significant differences in the ovarian and uterine sections of the uterine horn, in the middle part of the cervix, and in the hypothalamus were observed in group ZEN15 vs. groups ZEN5 and ZEN10. Groups ZEN5 and ZEN10 also differed in ZEN concentrations in the middle part of the cervix. On D2, significant differences were also found between group ZEN15 and groups ZEN5 and ZEN10, excluding both sections of the uterine horn in group ZEN5. On D3, significant differences were observed only in the ovaries, the ovarian section of the uterine horn, and the middle part of the cervix. 

The hierarchical analysis revealed that ZEN levels were highest on exposure date D1 in all analyzed tissues and in all groups. In all groups and on all exposure dates, the evaluated parameter was highest (see Appendix A) in the ovaries (proportional to the administered dose; Figure 1), followed by the middle part of the cervix, and, interestingly, the pituitary gland. 

Insignificant differences were noted between groups on D1, and between exposure dates in group ZEN5 (MABEL dose) (see Table 2). A comparison of α-ZEL levels in groups and on different exposure dates indicates that mean α-ZEL concentrations increased in proportion to the ZEN doses administered in groups, and to exposure dates, which probably could be attributed to the biotransformation of ZEN. The concentrations of the parent compound (see Table 1) in groups and on different exposure dates followed the opposite trend. 

In group ZEN10 (NOAEL dose) (see Table 2), significant differences in α-ZEL concentrations were observed in the ovaries and the uterine section of the uterine horn on D2 and D3, and in the middle part of the cervix on D3 relative to D1. Significant differences in the examined parameters were also noted in the uterine section of the uterine horn and in the middle part of the cervix between D2 and D3. In group ZEN15 (LOAEL dose) (see Table 2), α-ZEL levels in the ovaries and the hypothalamus differed significantly between D1 and D2. Significant differences in this parameter were found in the ovaries and in the ovarian section of the uterine horn between D1 and D3. In the ovarian section of the uterine horn, significant differences in α-ZEL concentrations were determined between D2 and D3. 

On D2 (see Table 2), significant differences in α-ZEL levels in the ovaries and the ovarian section of the uterine horn were observed between group ZEN5 and group ZEN15, and in the ovaries and the ovarian and uterine sections of the uterine horn between group ZEN5 and group ZEN10. On D3, significant differences in α-ZEL levels were noted only in the middle part of the cervix between group ZEN5 and groups ZEN10 and ZEN15. 

A graphic presentation of the CF values of α-ZEL (see Figure 2) revealed a certain trend: all CF values were inversely proportional to the results presented in Figure 1. The CF values of α-ZEL in the ovaries of group ZEN5 gilts on D1 were the only exception (the results were below the sensitivity of the method, and these CF values were expressed as 0). An analysis of ZEN and α-ZEL concentrations (see Table 1 and Table 2, respectively) on the remaining exposure dates in all groups revealed that α-ZEL levels were much lower than ZEN concentrations, and that the highest α-ZEL concentrations were noted in group ZEN5 on the last two exposure dates. The examined ZEN metabolite was present in nearly all hypothalamus samples. In the pituitary gland, the CF values of α-ZEL were very low in group ZEN15 on D2, and in groups ZEN10 and ZEN15 on D3 (see Figure 2), which indicates that α-ZEL was accumulated gradually in the pituitary gland and that its accumulation was inversely proportional to both the dose and exposure date. 

A hierarchical visualization of the CF values of α-ZEL in the examined tissues (see Appendix A) revealed several interesting findings. Firstly, α-ZEL was not detected in group ZEN5 on D1 (its levels were below the sensitivity threshold, see Table 2), and its concentrations peaked on D3. Secondly, α-ZEL concentrations in groups ZEN10 and ZEN15 were higher on D2 than on D1 and D3. Thirdly, the highest α-ZEL levels were noted in the middle part of the cervix, the ovaries, and both sections of the uterine horn, and its accumulation was highest (mathematically and hierarchically) in group ZEN5 on D2 and D3 (see Figure 2).

In group ZEN5, an absence of significant differences in β-ZEL concentrations was noted between tissues on different exposure days (see Table 3). In group ZEN10, clear differences in the studied parameter were found in the ovarian section of the uterine horn and in the middle part of the cervix between samples taken D1 and those taken on D2 and D3. In group ZEN15, differences in β-ZEL levels were noted between the ovaries and the uterine section of the uterine horn, but only on D3. 

No significant differences in β-ZEL concentrations were observed between groups on D1 and D2 (see Table 3). On D3, the CF (see Figure 3) values of β-ZEL in the ovaries, the ovarian section of the uterine horn, and the middle part of the cervix were lower in group ZEN5 than in groups ZEN10 and ZEN15. On D3, considerable differences in β-ZEL levels in the hypothalamus were also found between group ZEN5 and group ZEN15. The concentrations of β-ZEL in the middle part of the cervix and the hypothalamus differed significantly between group ZEN10 and group ZEN15.

The graphic presentation of the CF values of β-ZEL (see Figure 3) clearly indicates that the concentrations of this metabolite in the examined tissues were very low. Higher CF values that were proportional to the administered dose were noted only on D1 in groups ZEN10 and ZEN15. The ovaries were the only exception, where saturation with β-ZEL was higher than in the remaining tissues and not always proportional to the administered dose. In the pituitary gland, the CF values of β-ZEL were also higher than in the remaining tissues (excluding on D1), with the exception of the ovaries.

In group ZEN5, β-ZEL was not detected on D1 (see Appendix A) because the values determined in all tissues as below the sensitivity of the method (see Table 3 and Figure 3). In all groups, β-ZEL was not identified in the hypothalamus or pituitary gland on D1. On D2 and D3, the mathematical (see Table 3 and Figure 3) and hierarchical (see Appendix A) values of β-ZEL were generally low and similar. The CF values of β-ZEL were much higher only in the ovaries and the pituitary gland. The hierarchical order of data was not maintained between group ZEN10 and group ZEN15, but it was maintained between exposure dates (D1, D2, and D3).

The parent compound (ZEN) and its two metabolites (α-ZEL and β-ZEL) were deposited in reproductive system tissues, the hypothalamus, and the pituitary gland, even in gilts exposed to very low doses of ZEN (MABEL, NOAEL, and LOAEL). In the analyzed tissues, ZEN accumulation decreased proportionally on successive dates of exposure, and it decreased by ± 50% between D1 and D2, and between D2 and D3. 

#### 2.2.2. Blood Concentrations of estradiol and progesterone

Estradiol (E_2_) levels were higher in all experimental groups than in group C on all analytical dates (see Figure 4). However, significant differences were noted only on D3 and on successive exposure dates. On these dates, E_2_ concentrations were highest in group ZEN5 and lowest in group C (the differences were determined at 6.29 pg/mL on D3, 7.52 pg/mL on D4, 4.53 pg/mL on D5, and 4.96 pg/mL on D6). Estradiol levels were also higher in groups ZEN10 and ZEN15 than in group C on all exposure days, but the observed differences were not significant.

The distribution of P_4_ (progesterone) concentrations (see Figure 5) differed throughout the entire experiment. On the first four exposure dates, P_4_ levels were highest in group C. On D5 and D6, the analyzed parameter peaked in group ZEN10. Significant differences were observed between group ZEN10 and groups ZEN5 and C (on D5 0.21 ng/mL and D6 0.12 ng/mL).

## 3. Discussion

The results of this quantitative analysis of ZEN, α-ZEL, and β-ZEL levels in selected tissues of the reproductive tract and the HPG axis, and of E_2_ and P_4_ concentrations in the blood of pre-pubertal gilts may be difficult to interpret because very few studies have addressed this issue. 

On exposure date D1, the levels of ZEN and its two metabolites (see Table 1) in the examined tissues can be attributed to ongoing transformation processes in maturing gilts, which do not lead to hyperestrogenism, but merely cater to the physiological demand for endogenous steroids (see Figure 4 and Figure 5) in these animals [37,38]. This observation is problematic because physiologically maturing gilts (during puberty) are naturally deficient in endogenous steroids, and exposure to feed-borne ZEN leads to an increase in steroid levels. The question that arises is: how does this happen? One of the possible explanations is that pre-pubertal gilts adapt to frequent or prolonged exposure to low doses of exogenous steroid-like substances. The results noted for group ZEN5 (MABEL dose) on D1 seem to validate this hypothesis. 

We found a lack of both ZEN metabolites in the ZEN5 group at the D1 date (see Table 2 and Table 3), probably due to the low supply of endogenous steroid hormones and very low concentrations of exogenous (“free”) ZEN [37]. Exogenous ZEN was accumulated in large quantities in the studied tissues, including in the hypothalamus and pituitary gland (see Figure 1). The above could suggest that ZEN biotransformation in the intestines and blood vessels proceeded at a “slower” rate than ZEN binding to steroid hormone receptors, possibly to cater to a very high demand for steroids. This observation is supported by the CF values (see Figure 1) and the hierarchical order of data in the tree maps (see Appendix A), which suggest that ZEN was accumulated mostly in the ovaries and the cervix, followed by the pituitary gland, on D1 in group ZEN5 (MABEL dose) (see Appendix A). This process was accompanied by a marked increase in E_2_ concentrations (see Figure 4) in all experimental groups relative to the control group. On D3, E_2_ levels differed significantly between the experimental groups and the control group. At the same time, P_4_ concentrations (see Figure 5) were much lower in the experimental groups than in group C on all exposure days and remained at a similar level throughout the experiment. 

In groups ZEN10 and ZEN15 (NOAEL and LOAEL doses, respectively), ZEN concentrations in the hypothalamus and pituitary gland were inversely proportional to the administered doses, and a shift to other levels in the hierarchy was observed (see Appendix A, respectively). On D2 (see Appendix A), nervous system tissues ranked second in the data hierarchy. On D3 (see Appendix A), nervous system tissues ranked third in the data hierarchy only in group ZEN5 (MABEL dose). These results could point to the interdependence between the HPG axis and the ovaries, which is very important because ZEN exerts a direct negative effect on the neuroendocrine coordination of reproductive competence [39]. Similar observations were made by Rykaczewska et al. [10,37]. 

According to other researchers, α-ZEL is the dominant ZEN metabolite in the peripheral blood of pigs [17,40], and the results noted for group ZEN5 (MABEL dose) corroborate this observation. β-ZEL was the dominant ZEN metabolite in only one study [37]. The above could be explained by the host organism’s increased demand for compounds characterized by steroid (not only estrogenic) activity, such as α-ZEL and ZEN, but not β-ZEL [41]. These compounds are a source of endogenous steroid hormones which are essential for healthy functioning [42]. It should also be noted that ZEN and α-ZEL form highly stable complexes with albumins that prolong their half-life [7], which is a very important consideration in the interplay of steroid hormones. During exposure to ZEN, adaptive processes, in particular adaptive immunity, end on exposure date D1 [43]. Zearalenone is an endocrine-disrupting chemical (EDC) [44] and a substrate that regulates (inversely) the expression levels of genes encoding hydroxysteroid dehydrogenases (epigenetically [45]), which act as molecular switches and modulate steroid hormone pre-receptors [46]. Zearalenone also slows down proliferative processes in granulosa cells, proportionally to the administered dose, and provokes apoptosis in the ovaries [47]. These processes are accompanied by enterohepatic recirculation which slows down the elimination of mycotoxins [48]. Zearalenone was also found to affect gut microbiota [11] by increasing the log values of specific microbial counts in the distal segment of porcine intestines in the static and dynamic system. 

Much like other EDCs [30,44], ZEN can cross the blood-brain barrier, modify hypothalamic and pituitary functions, and exert a negative influence on central and peripheral reproductive tissues. According to He et al. [49], Rykaczewska et al. [37], and Zheng et al. [40], ZEN disrupts biological functions during the release of neurotransmitters which control physiological homeostasis, including levels of the FSH [33]. The discussed mycotoxin also blocks neuroactive ligand-receptor interaction pathways [39] and calcium signaling pathways [50,51,52] in the mitochondria. The resulting negative feedback initially decreases steroid production [45] and increases steroid synthesis on final dates of exposure. The results of this study could be also explained by the fact that ZEN blocks the activity of the HPG axis, which increases P_4_ levels, enhances endometrial receptivity, and induces specific morphometric changes in the reproductive system [37].

These observations suggest that the estrogen-sensitive tissues were saturated with ZEN (already on D1, proportionally to the administered ZEN dose), which leads to changes to the supraphysiological hormonal levels of pre-pubertal gilts, accompanied by an inverse correlation between steroid concentrations [37,38]. In pre-pubertal gilts exposed to ZEN, the strongest response was observed on D1, when hormonal receptors (not only estrogen receptors) were saturated, and neuroactive ligand-receptor interaction pathways were blocked. The accumulation of ZEN decreased considerably on successive dates of exposure, probably because the administered doses were very low, and the gilts’ physiological status perpetuated the processes that accompany ZEN mycotoxicosis. Extrapolation of the current findings indicates that ZEN, evaluated quantitatively in the analyzed tissues, affects HPG activity in pre-pubertal gilts in all examined doses due to its promiscuous properties [25].

Analysis of the concentrations (see Table 1, Table 2 and Table 3) and CF values (see Figure 1, Figure 2 and Figure 3 and Appendix A) of α-ZEL and β-ZEL based on arithmetic means and hierarchal data revealed certain trends. The concentrations and CF values of ZEN metabolites were inversely proportional to the corresponding values of the parent compound, in all tissues except the ovaries. In the examined nervous system tissues, ZEN metabolites accumulated gradually at concentrations that were inversely proportional to the administered dose (experimental group) and exposure date. This observation indirectly demonstrates that ZEN and α-ZEL (the most prominent mycoestrogens) exerted complementary effects and blocked estrogen receptors in the analyzed reproductive system tissues. However, ZEN probably slowed down the synthesis and release of FSH in the porcine pituitary gland [49]. Inhibited FSH secretion decreased the production of sex steroids, including E_2_, P_4_, and testosterone [34,37,45,53], and probably enhanced the conversion of steroids into E_2_, which increased feed intake and promoted the storage of excess energy in fat cells [10]. Therefore, inhibited pulsatile HPG activation slowed down gonadal development, prolonged sexual maturation, and delayed reproductive maturity.

A comparison of ZEN and α-ZEL concentrations (see Table 1 and Table 2) and the CF values of both mycotoxins, presented graphically, leads to another question: why in group ZEN5 (MABEL dose) was ZEN the dominant mycotoxin on D1, whereas α-ZEL was the dominant compound on D3? It could be postulated that “free ZEN” is initially used due to supraphysiological hormonal levels in pre-pubertal gilts [37], whereas biotransformation processes are initiated later, and excess “free” ZEN is metabolically transformed into, for example, α-ZEL which is bound to free estrogen receptors or used to block HPG activity. Therefore, both the parent compound and its metabolite act as epigenetic switches that regulate sexual maturation [45]. In the first week of exposure, only ZEN acts as an epigenetic switch, and on successive dates of exposure, both ZEN and α-ZEL play this role. The role of β-ZEL remains unknown.

## 4. Summary

The presented quantitative analysis suggests that ZEN and its metabolites can be among the factors that inhibit the somatic development of reproductive system tissues by decreasing the activity of the HPG axis in pre-pubertal gilts [23,34], which exerts a direct negative effect on the neuroendocrine coordination of reproductive competence [37,39]. However, from the breeders’ point of view, this is a positive phenomenon because low-dose ZEN mycotoxicosis enables maturing gilts to utilize nutrients for somatic development (most effectively in group ZEN5-MABEL dose) [10] rather than for reproductive development and performance. 

## 5. Materials and Methods

### 5.1. General Information

This article is a continuation of a previously published study protocol [54].

### 5.2. Experimental Feed

Analytical samples of ZEN were dissolved in 96 µl of 96% ethanol (SWW 2442-90, Polskie Odczynniki SA, Poland). Gilts were weighed at weekly intervals, and mycotoxin doses were calculated individually based on their BWs [6,11,12]. Gel capsules were saturated with the solution, feed was placed inside the capsules, and they were stored at room temperature to evaporate the alcohol. Throughout the trial, all gilts were given the same feed.

The feed given to all test animals was supplied by the same manufacturer. Friable feed was provided ad libitum twice everyday, at 8:00 a.m. and 5:00 p.m., during the experiment. The composition of the complete diet, as declared by the producer, is presented in Table 4 [6,11,12]. Pigs in the experimental groups received ZEN in gel capsules everyday before morning feeding. Feed was the carrier, and group C gilts received identical gel capsules without ZEN [10,11].

The approximate chemical composition of the diets fed to pigs in groups C, ZEN5, ZEN10, and ZEN15 was determined using the NIRS™ DS2500 F feed analyzer (FOSS, Hillerød, Denmark), a monochromator-based NIR reflection and transflectance analyzer (FOSS, Hillerød, Denmark) with a scanning range of 850–2500 nm [54].

#### Toxicological Analysis of Feed

Feed was analyzed for the presence of mycotoxins and their metabolites: ZEN, α-ZEL, and deoxynivalenol (DON). The level of the analyzed mycotoxins in feed samples were determined in accordance with the study protocol described previously [54]. Chromatographic methods were validated at the Department of Veterinary Prevention and Feed Hygiene, Faculty of Veterinary Medicine, University of Warmia and Mazury in Olsztyn, Olsztyn, Poland [55].

### 5.3. Experimental Animals

An in vivo experiment was performed at the Department of Veterinary Prevention and Feed Hygiene of the Faculty of Veterinary Medicine of the University of Warmia and Mazury in Olsztyn, Poland. The experiment involved 60 clinically healthy pre-pubertal gilts with an initial BW of 14.5 ± 2 kg [10,54]. During the experiment, the animals were housed in pens, fed identical diets, and had ad libitum access to water. The gilts were randomly divided into a control group (group C; *n* = 15) and 3 experimental groups (ZEN5, ZEN10, and ZEN15; *n* = 15 each). Groups ZEN5, ZEN10, and ZEN15 were administered ZEN (Sigma-Aldrich Z2125-26MG, St. Louis, MO, USA) per os at 5 µg/kg BW (MABEL dose), 10 µg/kg BW (NOAEL dose), and 15 µg/kg BW (LOAEL dose), respectively. Each experimental group was maintained in a separate pen in the same building. Each pen had an area of 25 m^2^, which complies with the applicable cross-compliance regulations (Regulation (EU) No 1306/2013 of the European Parliament and of the Council Brussels, Belgium of 17 December 2013). 

### 5.4. Toxicological Studies of Reproductive, Hypothalamic, and Pituitary Gland Tissues 

#### 5.4.1. Tissue Samples

Five prepubertal gilts from every group were euthanized on analytical date 1 (D1—exposure day 7), date 2 (D2—exposure day 21), and date 3 (D3—exposure day 42). Initially, general sedation was performed by intravenous administration of pentobarbital sodium (Fatro, Ozzano Emilia BO, Italy) and bleeding. Immediately after cardiac arrest, tissue samples (approximately 1 × 1.5 cm) were collected from the following segments: entire left ovary; left uterine horn (from the ovarian and uterine sections); middle part of the cervix; the entire hypothalamus; and the pituitary gland. The samples were rinsed with phosphate buffer and prepared for analyses. The collected samples were stored at a temperature of −20 °C.

#### 5.4.2. Extraction Procedure

The presence of ZEN, α-ZEL, and β-ZEL in tissue samples was determined in accordance with the with the study protocol described previously [54].

#### 5.4.3. Chromatographic Quantification of Zearalenone and Its Metabolites

Zearalenone and its metabolites were quantified at the Institute of Dairy Industry Innovation in Mrągowo, Poland. The biological activity of ZEN, α-ZEL, and β-ZEL in the tissues was determined in accordance with the study protocol described previously [54].

Mycotoxin concentrations were determined with an external standard and were expressed in ng/mL. Matrix-matched calibration standards were applied in the quantification process to eliminate matrix effects that can reduce sensitivity. Calibration standards were dissolved in matrix samples based on the procedure that was used to prepare the remaining samples. The material for the calibration standards was free of undesirable substances. The limits of detection (LOD) for ZEN, α-ZEL, and β-ZEL were determined as the concentration at which the signal-to-noise proportion decreased to 3. The concentrations of ZEN, alfa-ZEL and beta-ZEL were determined in each group and on 3 analytical dates (see Table 1).

#### 5.4.4. Mass Spectrometry Conditions

The electrospray ionization (ESI) mass spectrometer was operated in the negative ion operation. MS/MS parameters were optimized for every compound. Linearity was tested with a calibration curve including 6 levels. The optimal analytical conditions for the tested mycotoxins are presented in Table 5.

#### 5.4.5. CF

Carry-over toxicity takes place when an organism is able to survive under exposure to low doses of mycotoxins. Mycotoxins can exert a negative effect on tissues or organ function [56] and modify their activity [10,37]. The carry-over was determined in the examined tissues when the daily dose of zearalenone (5, 10, or 15 µg ZEN/kg BW) administered to each animal was equivalent to 560-32251.5 µg zearalenone/kg of the complete diet, depending on daily feed consumption. Mycotoxin contents in tissues were expressed in terms of the dry matter content of the samples. 

The CF was calculated as follows:CF = toxin concentration in tissue [ng/g]/toxin concentration in diet [ng/g] 

#### 5.4.6. Statistical Analysis

Data were processed statistically at the Department of Discrete Mathematics and Theoretical Computer Science, of the Faculty of Mathematics and Computer Science of the University of Warmia and Mazury in Olsztyn, Poland, as described previously [37]. 

### 5.5. Toxicological Studies of Blood

#### Blood samples collection

Blood was sampled in vivo from 5 gilts from every group on each analytical date. The first analytical date was exposure day 7 (D1); the second analytical date was exposure day 14 (D2); the third analytical date was exposure day 21 (D3); the fourth analytical date was exposure day 28 (D4); the fifth analytical date was exposure day 35 (D5), and the sixth analytical date was exposure day 42 (D6). Blood samples of 20 mL each were collected from all gilts (blood was sampled within 20 seconds after immobilization [57]) directly before slaughter by jugular venipuncture with a syringe containing 0.5 mL of heparin solution. The collected blood was centrifuged at 3000 rpm for 20 minutes at 4 °C. The obtained plasma samples were stored at −18 °C until estradiol (E_2_) and progesterone (P_4_) concentration analyses were performed. 

### 5.6. Determination of Plasma Hormone Concentrations 

#### 5.6.1. Estradiol

Estradiol concentrations in the blood plasma of gilts were analyzed at the Institute of Animal Reproduction and Food Research of the Polish Academy of Sciences in Olsztyn, Poland, as described previously [37]. Plasma E_2_ levels were determined by the radioimmunoassay (RIA) method with a commercial kit (ESTR-US-CT, CIS BIO ASSAYS) as described previously [58]. Estradiol concentrations were determined in accordance with the method described previously [37].

#### 5.6.2. Progesterone

Progesterone was quantified at the Analytical Laboratory of the Municipal Hospital with Polyclinic in Olsztyn, Poland, by the ECLIA electrochemiluminescence assay with the use of Elecsys Progesterone II and Cobas c 6000 analyzers (Hitachi, Tokyo, Japan). Progesterone concentrations were determined in accordance with the method described previously [37]. 

#### 5.6.3. Statistical Analysis

Data were processed statistically at the Department of Discrete Mathematics and Theoretical Computer Science, of the Faculty of Mathematics and Computer Science of the University of Warmia and Mazury in Olsztyn, Poland, as described previously [37]. 

## Figures and Tables

**Figure 1 toxins-14-00790-f001:**
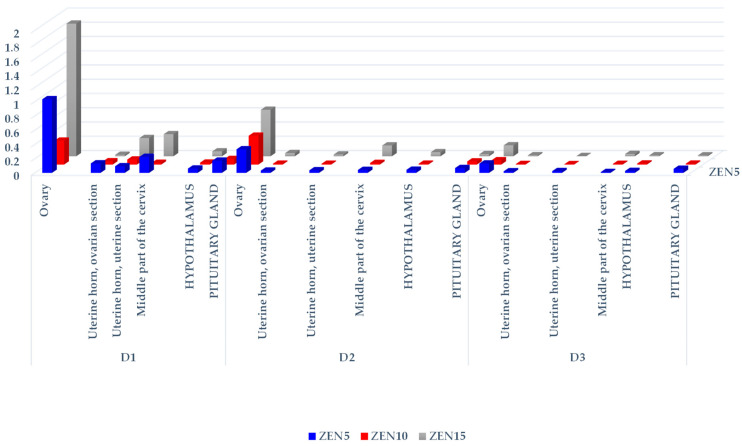
The CF of ZEN in the reproductive system tissues, hypothalamus, and pituitary glands of sexually immature gilts exposed to various ZEN doses. Key: D1—exposure day 7; D2—exposure day 21; D3—exposure day 42. Experimental groups: Group ZEN5—5 µg ZEN/kg BW; Group ZEN10—10 µg ZEN/kg BW; Group ZEN15—15 µg ZEN/kg BW.

**Figure 2 toxins-14-00790-f002:**
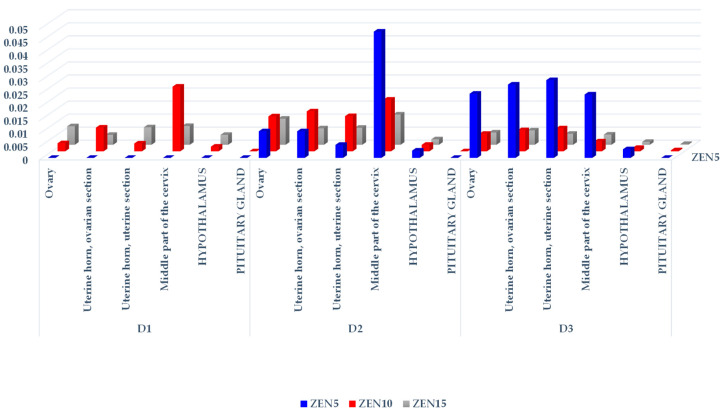
The CF of α-ZEL in the reproductive system tissues, hypothalamus, and pituitary glands of sexually immature gilts exposed to various ZEN doses. Key: D1—exposure day 7; D2—exposure day 21; D3—exposure day 42. Experimental groups: Group ZEN5—5 µg ZEN/kg BW; Group ZEN10—10 µg ZEN/kg BW; Group ZEN15—15 µg ZEN/kg BW.

**Figure 3 toxins-14-00790-f003:**
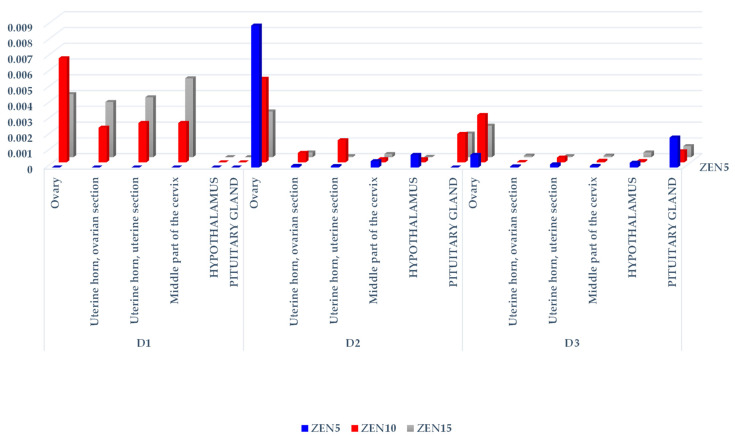
The CF of β-ZEL in the reproductive system tissues, hypothalamus, and pituitary glands of sexually immature gilts exposed to various ZEN doses. Key: D1—exposure day 7; D2—exposure day 21; D3—exposure day 42. Experimental groups: Group ZEN5—5 µg ZEN/kg BW; Group ZEN10—10 µg ZEN/kg BW; Group ZEN15—15 µg ZEN/kg BW.

**Figure 4 toxins-14-00790-f004:**
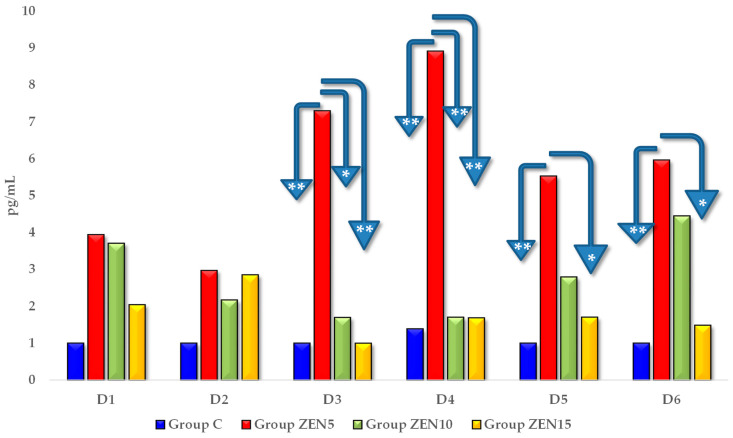
The effect of ZEN on blood E_2_ concentrations in pre-pubertal gilts: arithmetic means ($\bar{x}$) of five samples collected on each analytical date (D1–D6) in every group (control [C], ZEN5, ZEN10 and ZEN15). Statistics significant differences were determined at * *p* ≤ 0.05 and ** *p* ≤ 0.01.

**Figure 5 toxins-14-00790-f005:**
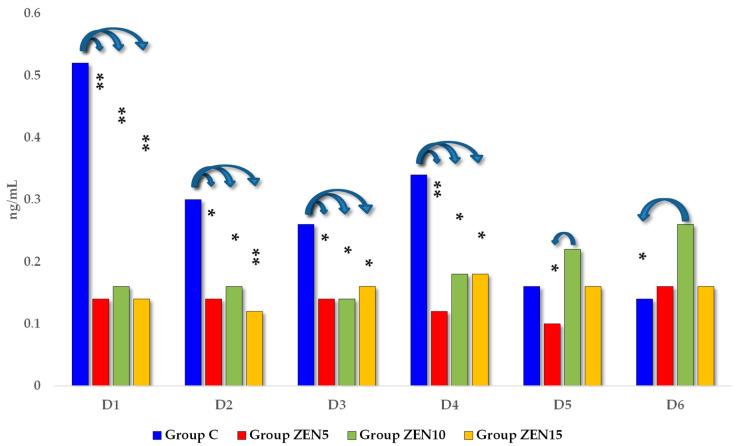
The effect of ZEN on blood P_4_ concentrations in pre-pubertal gilts: arithmetic means ($\bar{x}$) of five samples collected on each analytical date (D1–D6) in every group (control [C], ZEN5, ZEN10 and ZEN15). Statistics significant differences were determined at * *p* ≤ 0.05 and ** *p* ≤ 0.01.

**Table 1 toxins-14-00790-t001:** The CF and the mean (±) concentrations of ZEN (ng/g) in the reproductive system tissues, hypothalamus and pituitary glands of sexually immature gilts.

Exposure Dates	Feed Intake [kg/day]	Total ZEN Doses in Groups [µg/kg BW]	Tissue	Group ZEN5 [ng/g]	Group ZEN10 [ng/g]	Group ZEN15 [ng/g]
D1	0.8	80.5/161.9/242.7	Ovaries	83.15 ± 99.60	54.29 ± 51.37	451.67 ± 433.14
			Uterine horn, ovarian section	10.71 ± 5.13 ^xx^	7.46 ± 3.24 ^xx^	55.00 ± 17.00
			Uterine horn, uterine section	7.47 ± 1.52 ^x^	11.44 ± 5.24	62.08 ± 51.33
			Middle part of the cervix	18.06 ± 5.07 ^xx^	3.93 ± 0.93 ^xx yy^	75.19 ± 5.77
			Hypothalamus	4.88 ± 2.13 ^x^	4.43 ± 2.54 ^x^	16.76 ± 9.44
			Pituitary gland	13.77	12.66	14.26
D2	1.1	101.01/196.9/298.2	Ovary	33.40 ± 26.09	79.67 ± 19.94	194.56 ± 138.81
			Uterine horn, ovarian section	3.20 ± 3.32 ^a^	2.03 ± 0.88 ^a x^	13.40 ± 10.66 ^aa^
			Uterine horn, uterine section	3.67 ± 1.95 ^a^	2.20 ± 0.53 ^aa, x^	7.41 ± 4.60
			Middle part of the cervix	4.17 ± 3.11 ^aa xx^	4.46 ± 2.35 ^b^	45.35 ± 26.09
			Hypothalamus	4.39 ± 2.81 ^x^	2.38 ± 0.81 ^x^	17.07 ± 11.52
			Pituitary gland	6.81	8.63	9.02
D3	1.6	128.3/481.4/716.7	Ovary	16.80 ± 15.69 ^xx^	29.06 ± 17.79 ^xx^	110.38 ± 26.91
			Uterine horn, ovarian section	2.80 ± 1.71 ^a xx^	3.60 ± 1.84 ^x^	10.17 ± 2.37 ^aa^
			Uterine horn, uterine section	3.17 ± 1.80 ^aa^	2.73 ± 1.19 ^a^	2.47 ± 1.51
			Middle part of the cervix	1.51 ± 0.67 ^aa xx^	4.86 ± 2.89 ^xx^	23.17 ± 5.91 ^a^
			Hypothalamus	3.45 ± 1.20	6.93 ± 5.65	10.02 ± 2.72
			Pituitary gland	7.28	6.10	9.25

Abbreviations: D1—exposure day 7; D2—exposure day 21; D3—exposure day 42. Experimental groups: Group ZEN5—5 µg ZEN/kg BW; Group ZEN10—10 µg ZEN/kg BW; Group ZEN15—15 µg ZEN/kg BW. In the pituitary gland, ZEN concentrations were assayed in aggregate samples. The differences were regarded as statistically significant at ^a^, ^b^, ^x^
*p* ≤ 0.05 and ^aa^, ^xx^, ^yy^
*p* ≤ 0.01; ^a^, ^aa^ significant difference between exposure date D1 and exposure dates D2 and D3; ^x^, ^xx^ significant difference between group ZEN15and groups ZEN5 and ZEN10; ^yy^ significant difference between group ZEN5 and group ZEN10.

**Table 2 toxins-14-00790-t002:** The CF and the mean (±) concentrations of α-ZEL (ng/g) in the reproductive system tissues, hypothalamus, and pituitary glands of sexually immature gilts.

Exposure Dates	Feed Intake [kg/day]	Total ZEN Doses in Groups [µg/kg BW]	Tissue	Group ZEN5 [ng/g]	Group ZEN10 [ng/g]	Group ZEN15 [ng/g]
D1	0.8	80.5/161.9/242.7	Ovary	0	0.51 ± 0.34	1.73 ± 0.45
			Uterine horn, ovarian section	0	1.48 ± 0.98	0.94 ± 0.25
			Uterine horn, uterine section	0	0.50 ± 0.28	1.65 ± 0.18
			Middle part of the cervix	0	4.03 ± 0.16	1.77 ± 1.46
			Hypothalamus	0	0.30 ± 0.20	0.94 ± 0.04
			Pituitary gland	0	0	0
D2	1.1	101.01/196.9/298.2	Ovary	1.04 ± 0.10 ^xx yy^	2.65 ± 0.55 ^aa^	3.00 ± 0.04 ^a^
			Uterine horn, ovarian section	1.04 ± 0.42 ^yy^	3.02 ± 0.19	1.90 ± 0.87
			Uterine horn, uterine section	0.51 ± 0.22 ^xx yy^	2.67 ± 0.42 ^aa^	1.95 ± 0.48
			Middle part of the cervix	4.89 ± 0.60 ^x^	3.91 ± 0.08	3.48 ± 0.48
			Hypothalamus	0.29 ± 0.20 ^x^	0.50 ± 0.03	0.65 ± 0.08 ^a^
			Pituitary gland	0	0	0.246
D3	1.6	128.3/481.4/716.7	Ovary	3.16 ± 0.73	3.23 ± 0.38 ^aa^	3.38 ± 0.54 ^aa^
			Uterine horn, ovarian section	3.61 ± 0.16	3.91 ± 1.66	4.00 ± 0.32 ^aa, bb^
			Uterine horn, uterine section	3.83 ± 0.17	4.24 ± 0.47 ^aa bb^	3.02 ± 1.80
			Middle part of the cervix	3.13 ± 0.19 ^xx yy^	1.83 ± 0.17 ^aa bb^	2.81 ± 0.27
			Hypothalamus	0.43 ± 0.29 ^x^	0.65 ± 0.08	0.79 ± 0.15
			Pituitary gland	0	0.195	0.245

Abbreviations: D1—exposure day 7; D2—exposure day 21; D3—exposure day 42. Experimental groups: Group ZEN5—5 µg ZEN/kg BW; Group ZEN10—10 µg ZEN/kg BW; Group ZEN15—15 µg ZEN/kg BW. LOD > values below the limit of detection were expressed as 0. In the pituitary gland, α-ZEL concentrations were assayed in aggregate samples. The differences were regarded as statistically significant at ^a^, ^x^
*p* ≤ 0.05 and ^aa^, ^bb^, ^xx^, ^yy^
*p* ≤ 0.01; ^a^, ^aa^ significant difference between exposure date D1 and exposure dates D2 and D3; ^b^, ^bb^ significant difference between exposure date D2 and exposure date D3; ^x^, ^xx^ significant difference between group ZEN15 and groups ZEN5 and ZEN10; ^yy^ significant difference between group ZEN5 and group ZEN10.

**Table 3 toxins-14-00790-t003:** The CF and the mean (±) concentrations of β-ZEL (ng/g) in the reproductive system tissues, hypothalamus, and pituitary glands of sexually immature gilts.

Exposure Date	Feed Intake [kg/day]	Total ZEN Doses in Groups [µg/kg BW]	Tissue	Group ZEN5 [ng/g]	Group ZEN10 [ng/g]	Group ZEN15 [ng/g]
D1	0.8	80.5/161.9/242.7	Ovary	0	1.07 ± 0.29	0.98 ± 0.12
			Uterine horn, ovarian section	0	0.36 ± 0.07	0.87 ± 0.14
			Uterine horn, uterine section	0	0.41 ± 0.08	0.93 ± 0.03
			Middle part of the cervix	0	0.42 ± 0.04	1.22 ± 0.02
			Hypothalamus	0	0	0
			Pituitary gland	0	0	0
D2	1.1	101.01/196.9/298.2	Ovary	0.92 ± 0.10	1.05 ± 0.53	0.87 ± 0.06 ^xx^
			Uterine horn, ovarian section	0.02 ± 0.02	0.12 ± 0.08 ^aa^	0.09 ± 0.01 ^aa^
			Uterine horn, uterine section	0.01 ± 0.01	0.29 ± 0.57	0.02 ± 0.02 ^aa^
			Middle part of the cervix	0.05 ± 0.01	0.04 ± 0.01 ^aa^	0.06 ± 0.02 ^aa^
			Hypothalamus	0.09 ± 0.13	0.04 ± 0.009	0.006 ± 0.01
			Pituitary gland	0	0.372	0.462
D3	1.6	128.3/481.4/716.7	Ovary	0.11 ± 0.12 ^xx yy^	1.48 ± 0.25	1.49 ± 0.27 ^a^
			Uterine horn, ovarian section	0.01 ± 0.005 ^xx yy^	0.01 ± 0.002 ^aa^	0.14 ± 0.02 ^aa^
			Uterine horn, uterine section	0.03 ± 0.01	0.14 ± 0.18	0.05 ± 0.03 ^aa^
			Middle part of the cervix	0.02 ± 0.01 ^x yy^	0.07 ± 0.02 ^aa z^	0.12 ± 0.02 ^aa^
			Hypothalamus	0.04 ± 0.03 ^yy^	0.04 ± 0.01 ^zz^	0.21 ± 0.03
			Pituitary gland	0.253	0.362	0.536

Abbreviations: D1—exposure day 7; D2—exposure day 21; D3—exposure day 42. Experimental groups: Group ZEN5—5 µg ZEN/kg BW; Group ZEN10—10 µg ZEN/kg BW; Group ZEN15—15 µg ZEN/kg BW. LOD > values below the limit of detection were expressed as 0. In the pituitary gland, β-ZEL concentrations were assayed in aggregate samples. The differences were regarded as statistically significant at ^a^, ^x^, ^z^
*p* ≤ 0.05 and at ^aa^, ^xx^,^**yy**^, ^zz^
*p* ≤ 0.01; ^a^, ^aa^ significant difference between exposure date D1 and exposure dates D2 and D3; ^x^, ^xx^ significant difference between group ZEN5 and group ZEN10; ^yy^ significant difference between group ZEN5 and group ZEN15; ^z^, ^zz^ significant difference between group ZEN10 and group ZEN15.

**Table 4 toxins-14-00790-t004:** Declared composition of the complete diets [37].

Parameters	Composition Declared by The Manufacturer (%)
Soybean meal	16
Wheat	55
Barley	22
Wheat bran	4.0
Limestone	0.3
Zitrosan	0.2
Vitamin-mineral premix ^1^	2.5

^1^ Composition of the vitamin-mineral premix per kg: vitamin A—500,000 IU; iron—5000 mg; vitamin D3—100,000 IU; zinc—5000 mg; vitamin E (alpha-tocopherol)—2000 mg; manganese—3000 mg; vitamin K—150 mg; copper (CuSO4·5H2O)—500 mg; vitamin B1—100 mg; cobalt—20 mg; vitamin B2 —300 mg; iodine—40 mg; vitamin B6—150 mg; selenium—15 mg; vitamin B12—1500 μg; L-lysine—9.4 g; niacin—1200 mg; DL-methionine + cystine—3.7 g; pantothenic acid—600 mg; L-threonine—2.3 g; folic acid—50 mg; tryptophan—1.1 g; biotin—7500 μg; phytase + choline—10 g; ToyoCerin probiotic+calcium—250 g; antioxidant+mineral phosphorus and released phosphorus—60 g; magnesium—5 g; sodium and calcium—51 g.

**Table 5 toxins-14-00790-t005:** Optimized conditions for the tested mycotoxins [15].

Analyte	Precursor	Quantification Ion	Confirmation	Ion LOD (ng mL^−1^)	LOQ (ng mL^−1^)	Linearity (%R^2^)
ZEN	317.1	273.3	187.1	0.03	0.1	0.999
α-ZEL	319.2	275.2	160.1	0.3	0.9	0.997
β-ZEL	319.2	275.2	160.1	0.3	1	0.993

## Data Availability

Not applicable.

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
