# Peer review of "Exposure to Low Zearalenone Doses and Changes in the Homeostasis and Concentrations of Endogenous Hormones in Selected Steroid-Sensitive Tissues in Pre-Pubertal Gilts"

_toxins, 2022, doi:10.3390/toxins14110790_

Round 1

Reviewer 1 Report

This study was carried out to determine whether prolonged exposure to low-dose zearalenone (ZEN) affects the concentrations of ZEN, α-zearalenol (α-ZEL), and β-zearalenol (β-ZEL) in selected tissues, or the concentrations of selected steroid hormones in pre-pubertal gilts. The manuscript was well prepared and supplied enough data to support its conclusions. I suggested it could be accepted after minor revision.

minor revision

1.Table 1, Table 2, Table 3 is not easy to read. Considering the complexity of multiple factors such as toxin exposure level, exposure time and tissue site,I suggest you split concentrations and carry over factor into different tables.

2. In the above tables, the superscripts of significance difference were confusing. I suggest you may use superscripts such as a.b.c to compare the main effects of ZEN dose, while superscripts such as x,y,z to compare the main effects of exposure time. The p value should be supplemented in the tables.

3. Figure 1, Fig 2 and Fig 3 didn't supply more information than the tables. I suggest to delete these 3 figures.

4.L394 and L399. Statement repetition. please make them more concise. 

5.L522, about the CF calculation. The CF of ZEN is easy to be understood. But did the CF of α-ZEL, and β-ZEL equal to the concentration of α-ZEL or β-ZEL in tissue [ng/g] / ZEN concentration in diet [ng/g]?

Author Response

Dear Reviewer,

First of all, we would like to express our gratitude for reading and suggesting improvements to the manuscript. All of the points you raised were carefully analyzed and addressed. Below please find detailed replies to your comments.

All corrections introduced in the manuscript are highlighted in green.

We hope that in the present form the manuscript fulfills your expectations and can be accepted for the publication.

1.Table 1, Table 2, Table 3 is not easy to read. Considering the complexity of multiple factors such as toxin exposure level, exposure time and tissue site,I suggest you split concentrations and carry over factor into different tables.

Ad 1. – We agree with the Reviewer and have reworded all three tables according to his suggestions;

  1. In the above tables, the superscripts of significance difference were confusing. I suggest you may use superscripts such as a.b.c to compare the main effects of ZEN dose, while superscripts such as x,y,z to compare the main effects of exposure time. The p value should be supplemented in the tables.

Ad 2. – As in point 1. We made corrections and changed the superscripts, which makes them easier to read. P-values are presented everywhere;     

  1. Figure 1, Fig 2 and Fig 3 didn't supply more information than the tables. I suggest to delete these 3 figures.

Ad 3. – Taking advantage of the opportunity that we have eliminated the carry over factor values in the tables, we propose not to present them tabularly but to leave them in the form of figures. This is due to two reasons – (i) there are no statistical differences; (ii) the proposed visual form in the form of Figures is more readable and perfectly represents the very variable metabolic situation of these low doses of ZEN and its metabolites.

4.L394 and L399. Statement repetition. please make them more concise. 

Ad 4. – This second fragment (L399) has been removed;

5.L522, about the CF calculation. The CF of ZEN is easy to be understood. But did the CF of α-ZEL, and β-ZEL equal to the concentration of α-ZEL or β-ZEL in tissue [ng/g] / ZEN concentration in diet [ng/g]?

Ad 5. – Yes, because metabolites are the result of biotransformation of the parent substance. In other words, they are the part of ZEN that has not reacted with all sorts of receptors, directly with tissues, specific microbiotas, or has not been excreted in urine or feces. Answering the second part of the question, both metabolites were determined in tissues in ng/g, while in feed they were determined in μg/kg and then converted in the 1:1000 system due to the fact that 1 μ = 1000 ng. The form of conversion resulting from the used feed is presented in L516-520.

Reviewer 2 Report

Review for

 Article 

Exposure to Low Zearalenone Doses and Changes in the Home-2 ostasis and Concentrations of Endogenous Hormones in Se-3 lected Steroid-Sensitive Tissues in Pre-pubertal Gilts

As mycotoxins represent a major threat for animals and humans at a worldwide level, it is important to strengthen research efforts in this area

The present paper investigates prolonged exposure to low-dose zearalenone (ZEN) mycotoxicosis

this study determined whether prolonged exposure to low-dose zearalenone (ZEN) mycotoxicosis affects the concentrations of ZEN, α-zearalenol (α-ZEL), and β-zearalenol (β-ZEL) in selected reproductive system tissues (ovaries, uterine horn – ovarian and uterine sections, and the middle part of the cervix), the hypothalamus and pituitary gland, or the concentrations of selected steroid hormones in pre-pubertal gilts.

Very good paper. Experiments well conducted. Interpretation and discussion accompanied by adequate results.

 The results presented in this work suggest that ZEN, α-ZEL, and β-ZEL were deposited in all analyzed tissues despite exposure to very low ZEN doses. The presence of these undesirable compounds in the examined tissues can inhibit the somatic development of the reproductive system and compromise neuroendocrine coordination of reproductive competence in pre-pubertal gilts.

The presented results were obtained as part of a large-scale experiment which did not reveal clinical signs of ZEN mycotoxicosis.

Here we are speaking about ‘low-noise’ effects, which have ALSO a strong importance.

However, from the breeders’ point of view, this is a positive phenomenon because low-dose ZEN mycotoxicosis enables maturing gilts to utilize nutrients for somatic development (most effectively in group ZEN5 – MABEL dose) [10] rather than for reproductive development and performance.

Is it a true and good reason to allow low-dose ZEN mycotoxicosis?  Or low-dose ZEN mycotoxicosis and any-dose ZEN mycotoxicosis should be avoided at any time?

Author Response

Dear Reviewer,

First of all, we would like to express our gratitude for reading our manuscript. The issue you raised has been carefully analyzed and resolved. Below you will find a detailed answer to your attention.

We hope that the manuscript in its current form will meet your expectations and can be accepted for publication.

 As mycotoxins represent a major threat for animals and humans at a worldwide level, it is important to strengthen research efforts in this area

 The present paper investigates prolonged exposure to low-dose zearalenone (ZEN) mycotoxicosis

 this study determined whether prolonged exposure to low-dose zearalenone (ZEN) mycotoxicosis affects the concentrations of ZEN, α-zearalenol (α-ZEL), and β-zearalenol (β-ZEL) in selected reproductive system tissues (ovaries, uterine horn – ovarian and uterine sections, and the middle part of the cervix), the hypothalamus and pituitary gland, or the concentrations of selected steroid hormones in pre-pubertal gilts.

 Very good paper. Experiments well conducted. Interpretation and discussion accompanied by adequate results.

 The results presented in this work suggest that ZEN, α-ZEL, and β-ZEL were deposited in all analyzed tissues despite exposure to very low ZEN doses. The presence of these undesirable compounds in the examined tissues can inhibit the somatic development of the reproductive system and compromise neuroendocrine coordination of reproductive competence in pre-pubertal gilts.

 The presented results were obtained as part of a large-scale experiment which did not reveal clinical signs of ZEN mycotoxicosis.

Here we are speaking about ‘low-noise’ effects, which have ALSO a strong importance.

 However, from the breeders’ point of view, this is a positive phenomenon because low-dose ZEN mycotoxicosis enables maturing gilts to utilize nutrients for somatic development (most effectively in group ZEN5 – MABEL dose) [10] rather than for reproductive development and performance.

Is it a true and good reason to allow low-dose ZEN mycotoxicosis?  Or low-dose ZEN mycotoxicosis and any-dose ZEN mycotoxicosis should be avoided at any time?

Dear Reviewer

Thank you very much for taking the time to read our work and for your very positive comments.

Answering the problem question, we would like to note that the widespread use of generally available detoxes is probably more dangerous to health than these small doses of mycotoxins. On the other hand, tolerating the dose of MABEL should not cause anyone health problems because our many years of research show that animal organisms do not show any clinical signs of disease due to its presence in the feed. However, it is advisable to decide whether to introduce a detoxified feed additive  or not, depending on the value of e.g. zearalenone, as  a result of testing feed materials for the presence of  mycotoxins before the production of feed, depending on the value of e.g. zearalenone?

Reviewer 3 Report

A rather interesting study, including quantitative analysis of ZEN, α-ZEL, and β-ZEL levels in selected tissues of the reproductive tract and the HPG axis, and E2 and P4 concentrations in the blood of pre-pubertal gilts, that comes as a sequence of other published material. The manuscript has been submitted to a special issue where previous articles that included results from the same main study are already presented. Particular moderations are needed as suggested in comments of attached pdf file.

Author Response

Dear Reviewer,

Thank you very much for your time and careful reading of the manuscript. We greatly appreciate your comments and believe that the revision of the manuscript according to your suggestions will enhance its quality and make it acceptable for the publication.

All corrections introduced in the manuscript are highlighted in yellow.

Please find our replies to your comments:

Popup note - CF should be defined at first appearance in the text.

L 95 – carry over factor completed;

Popup note – L 114-116 – In comparison to D1?

L 114-116 – the marked sentence is supplemented by the entry "in comparison to D1";

Popup note – L 153 – Insignificant

L 153 – the word "no significant" is changed to "insignificant";

Popup note – L 186-187 – That's an issue for the discussion section.

L 186-187 – this is also a fact. In our opinion, the facts can (and even should) be presented in the results chapter;

Popup note – L 196-199 – Same as the previous comment for the discussion section. Topographical cellular differences, amount and type of estrogen receptors, cascade of receptors  binding events, adaptation mechanisms of gilts, under the prism of the lowest ZEN dose used, should be discussed.

L 196-199 – these are also "facts" and we propose to leave them in the text;

Popup note – L 211 – absence of significant

L 211 – the word "no" was deleted and the words "absence of" were introduced;

Popup note – L 263 – on which day?

L 263 – The terms D5 and D6 were introduced;

Popup note – L 268 – The ZEN5 group effect on E2 only after D3 and the absence of  similar effect in ZEN15 group need further discussion. Quite similarly increasing P4 levels on Days 5 & 6 in ZEN10 group but not in ZEN15 groups still remain a small mystery.

L 268 – Dear Reviewer, already in the first paragraph we state that the concentrations of tested hormones in the blood during experimental zearalenone mycotoxicosis are difficult due to the lack of knowledge or ability to interpret the situation. We continue to maintain this position. As much as we think is a sensible translation, we have presented so much in this chapter;

Popup note – L 283 – Where does this refer to?

L 283 – The beginning of the sentence has been redrafted and supplemented – We found a lack of both ZEN metabolites in the ZEN5 group at the D1 date…..;

Popup note – L 360-361 – Is that finding observed in the present study?

L 360-361 – Of course not. This finding was documented in an earlier publication to which we refer at the end of the sentence [10].;

Popup note – L 416 – The feed wasn't analyzed for other major mycotoxins such as AFB1, OTA and FBs?

L 416 – Prevalence studies  show that the probability of occurrence of these mycotoxins is small. As well as we used feed produced by a local producer who buys only native cereals for the production of feed;

Popup note – L 448 – Was there a number of replicates in each group? All animals per group were in one pen?

L 448 – We answer both questions yes. This follows from the text in L 448-451;

Popup note – L 457 – Was there any sedation used prior to pentobarbital?

L 457 - “Initially general sedation was performed, and then intravenously administered pentobarbital sodium” - such, the information was introduced in the text;

Popup note – L 523 – Taking into account that the levels of ZEN used were minimal, was there a sample size calculation performed in order to verify any extrapolation of results?

L 523 – No calculation of the sample size was carried out to verify the extrapolation of the results. The choice of doses used in our study is specified in an article published in EFSA Journal – July 2017 DOI: 10.2903/j.efsa.2017.4851 Risks for animal health related to the presence of zearalenone and its modified forms in feed – written by 26 authors.  It determines the NOAEL dose for zearalenone, which is the basis for determining the size of the other two samples;

Popup note – L 532 – The one way ANOVA was only used in repeated measurements of residues? L 594 – Again the question is if a one way ANOVA is capable of interpreting differences in repeated measurements of the same parameter?

L 532 and 594 – Yes, because samples of one tissue (e.g. ovary) or blood for one hormone were taken in one experiment group from five gilts at a specific exposure date. The results obtained from one tissue were compared with other experimental groups at different exposure dates;

Popup note – L 546 – selected on which basis?

L 546 – based on each random selection of gilts to be downloaded;

Popup note – L 567 – 5% in both cases?

L 567 – Yes in both cases;

Popup note – L 601 – Link not functional

L 601 – This link is launched by the publisher;

Popup note - L 603 – no question;

Popup note – L 604 – Ethical Committee approval details should be inserted.

L 604 – In the original sent to the Editorial Board, everything was given. We are reintroducing;

Popup note - L 605 – no question.
